# A Higher Healthy Eating Index Is Associated with Decreased Markers of Inflammation and Lower Odds for Being Overweight/Obese Based on a Case-Control Study

**DOI:** 10.3390/nu14235127

**Published:** 2022-12-02

**Authors:** Farhad Vahid, Mahsa Jalili, Wena Rahmani, Zahra Nasiri, Torsten Bohn

**Affiliations:** 1Nutrition and Health Research Group, Precision Health Department, Luxembourg Institute of Health, 1445 Strassen, Luxembourg; 2Preventive and Clinical Nutrition Group, Department of Nutrition, Exercise and Sports, Faculty of Science, University of Copenhagen, 2200 Copenhagen, Denmark; 3School of Medicine, Arak University of Medical Sciences, Arak 3819693345, Iran

**Keywords:** reactive oxygen species, immune system, antioxidants, cardiometabolic disease, cytokines

## Abstract

Obesity is a health risk characterized by chronic inflammation, and food choices are strongly associated with its etiology. Our objective was to investigate the association between dietary patterns and the healthy eating index (HEI) with the odds of overweight/obesity and related inflammatory markers. Within a population-based case-control study, we collected data and samples from 793 normal-weight and 812 overweight/obese Iranian people (based on either body mass index (BMI) or body surface area (BSA)). Dietary intake and HEI scores were obtained via a validated 124-item food frequency questionnaire. Anthropometric and socioeconomic parameters, as well as blood inflammatory markers, were measured. Participants with higher HEI scores had higher serum high-density lipoprotein-cholesterol (HDL-C) and significantly lower energy intake. Water consumption in the overweight/obese group was significantly lower than in the control group. In the final models using partial correlation and controlling for multiple confounders, there was a significant inverse correlation between HEI and interleukin-4 (IL-4, R = −0.063), IL-1β (R = −0.054), and high-sensitivity C-reactive protein (hs-CRP, R = −0.069). Based on multivariable logistic regression models adjusted for multiple confounders, there was a significant association between HEI as a continuous variable (OR = 0.993, 95% CI: 0.988–0.999) and categorical variable (OR = 0.801, 95% CI: 0.658–0.977) and odds of overweight/obesity across BMI groups. The dietary patterns in the case and control groups however were similar, and we failed to find a significant association between HEI and odds of overweight/obesity based on BSA. Adherence to healthy eating recommendations may be a prudent recommendation to prevent overweight/obesity and keeping inflammatory indicators low.

## 1. Introduction

Obesity is a prevalent epidemic resulting from excessive or abnormal accumulation of body fat mass, especially visceral fat, and is a risk factor for several especially non-communicable diseases, namely cardiovascular diseases (CVD), type 2 diabetes (T2D), and cancer, posing a large psychological and physical burden on the affected individuals [1,2]. There are different definitions for obesity, but the most common one is based on body mass index (BMI) as a person’s weight (in kg) divided by the square of their height (in m) [3]. Although the gold standard for measuring body fat is under water weighing (UWW) [4], some accurate alternative techniques, such as dual-energy X-ray absorptiometry (DEXA) and bioelectrical impedance analysis (BIA), have been used to more practically assess body composition and body fat percentage [5]. According to the World Health Organization (WHO), overweight in adults is a BMI of 25 to ˂30 kg/m^2^, and obesity is a BMI greater than or equal to 30 [6] kg/m^2^. In addition to BMI, there are other more refined methods to define obesity, such as based on the body surface [7].

The prevalence of obesity is on the rise in many countries, and it has become a global pandemic that has reduced the life expectancy of affected populations. According to a published report in 2017, 4.2 million deaths worldwide have been related to being overweight and obese. In other words, of all death records, 8% were linked to obesity, and 15% were related to overweight [8].

Between 1975 and 2016, the obesity rate reached epidemic proportions, and the prevalence has increased by three times [9]. According to a WHO report in 2016, over 1.9 billion adults worldwide were classified as overweight, and over 650 million (11% of men and 15% of women) were obese [9]. This is not limited to Western countries. In Iran, the percentage of adults with overweight and obesity has been reported as 59.3% and 22.7%, respectively [10]. It is expected that around 40% of the world’s population will be overweight (BMI of 25 to <30 kg/m^2^) and class 1 obese (BMI of 30 to <35 kg/m^2^) by 2030 [2]. Obesity is associated with a higher risk for many co-morbidities, including hyperlipidemia, insulin resistance, osteoarthritis, sleep apnea, polycystic ovary syndrome (PCO), depression, and high blood pressure [1]. In addition, the odds for T2D are 10 times greater in persons with obesity, and 20–30% of deaths in patients with CVD are related to overweight and obesity [1]. Moreover, obesity stigma and discrimination based on weight are widespread in society [11]. In terms of economic prospects, direct costs of obesity and overweight in the health care system are estimated at 2 to 6% of the gross domestic income, and, in developed countries, the burden of costs is much higher [1].

Taking into account the importance and potential consequences of obesity, the American Medical Association has introduced obesity as a separate disease [2]. The leading causes of obesity are excess caloric intake, lifestyle changes, excess intake of processed foods and simple carbohydrates, reduced physical activity (i.e., a sedentary lifestyle), automation of everyday life, and unhealthy eating patterns [2,6]. Among environmental factors, dietary patterns and diet quality have been recognized as the main causes [12,13,14].

The healthy eating index (HEI)—a measure to assess how well a set of foods align with dietary recommendations—is a valid approach to examining the quality of dietary intake and nutritional assessment based on healthy eating recommendations for Americans [15,16,17] and can indicate an association between dietary intake and risk of diseases. For example, Western dietary patterns have been associated with higher BMI and waist-to-hip ratio (WHR). In contrast, people with prudent dietary patterns, including a higher intake of vegetables, seafood, and non-hydrogenated fats, tend to have a lower BMI and WHR [18].

The association between HEI and obesity has been investigated in some studies, and healthy dietary intakes that follow HEI recommendations were shown to be associated with a lower risk of being overweight or obese [19]. However, very limited studies have investigated the association between HEI and obesity in Iran. Since the dietary patterns in developing countries such as Iran are moving toward a Western dietary pattern [20,21,22], a study considering factors in the dietary patterns/habits can be useful to overcome the limitations of previous studies. Specifically, most of the studies were conducted as cross-sectional sampling and the data could not be generalized for long-term dietary quality, and the evidence for drawing a conclusion on a cause–effect association is too limited [20,21,23]. According to our knowledge, few studies have considered both dietary patterns and HEI and serum inflammatory profile; therefore, our study may be of great interest in this field. Our main objective was to investigate the association of dietary patterns and HEI scores with the odds of overweight/obesity in Iranian adults, including body surface area (BSA) as a rarely investigated marker of overweight/obesity compared to BMI. In addition, this study can be a valuable example to compare the association between traditional and Western dietary patterns and risk of obesity.

## 2. Materials and Methods

### 2.1. Participants

The full protocol of the study, including methods and materials, has been published elsewhere [24]. Briefly, this population-based case-control study was conducted in Arak Medical Centers, Arak, Iran. The cases and controls were recruited according to the BMI criteria. The cases were 812 overweight/obese participants (BMI equal to or greater than 25 kg/m^2^), and the controls were 793 normal-weight participants (BMI between 17.5 and 24.9 kg/m^2^). The eligible volunteers interested in participating in this study were selected after an interview, explaining research procedures and measuring body weight and height. Afterward, the study questionnaires were filled out, and venous blood samples were taken for biochemical analysis. Written informed consent was received from all participants in this study, and the Arak University of Medical Science Ethics Committee, Arak, Iran approved the study protocol (Ethics Committee No. IR. ARAKMU. REC.1398.094).

The inclusion criteria were as follows: age range 18–81 years old; interested in participation in the study; filling out at least 80% of the questionnaire; signed written informed consent; living in Arak for at least the past 5 years. The exclusion criteria were as follows: current active diseases, such as CVD, liver and kidney diseases, and cancer; taking any medication (especially anti-inflammatory drugs) and supplements; drug addiction; being on a special diet, such as the vegetarian or weight loss diet; any major change in dietary intake during the last 6 months; pregnancy or lactation.

### 2.2. Assessment of Blood Parameters

From all individuals, five milliliters of intravenous blood were obtained after 8–12 h of fasting. Serum was obtained from whole blood by centrifugation at 2000× *g* for 10 min. Serum samples were then collected and stored at −80 °C until further analysis. The following methods measured serum inflammatory and antioxidant biomarkers by approved laboratories. Hs-CRP levels were measured by the hypersensitivity method using an ELISA kit (hs-CRP kit provided by DRG International Inc., Springfield, NJ, USA). The minimum detectable concentration of the abovementioned kit was about 0.1 mg/L. As recommended by the American Heart Association, the cut-off level for inflammation regarding hs-CRP was 3 mg/dL. According to the manufacturer’s instructions, serum levels of ILs and TNF-α were assessed using relevant commercial ELISA kits (ab181421, ABCAM; Trumpington, Cambridge, UK). MDA and TAC assay kits (Teb Pazhouhan Razi, Tehran, Iran) were used to measure serum antioxidant capacity. Rapid insulin ELISA kit (Monoband) and hemoglobin A1C (HbA1c) enzymatic kit (Pishtazteb, Iran) were used to measure biochemical markers of glucose metabolism.

### 2.3. Assessment of Anthropometric and Socioeconomic Variables

A Seca scale measured the participant’s weight with an accuracy of 50 g, and height was assessed using a tape measure attached to the wall with an accuracy of 1 cm. The BMI was considered “normal” if the value was between 17.9 and 24.9 kg/m^2^ and obese/overweight if the value was equal to or higher than 25 kg/m^2^. The BSA was calculated according to the Mosteller formula: BSA = (height (cm) × weight (kg)/(3600 kg))^1/2^ [24]. The BSA was considered “normal” if the mean value was ≤1.91 m^2^ for men and ≤1.71 m^2^ for women.

A general information questionnaire gathered information on gender (female/male), education (lower than diploma/higher than diploma), marital status (single/married/widowed, divorced, separated, not willing to mention, living with a partner), smoking (yes/no), alcohol (yes/no), regular physical activity (yes/no), history of CVD (yes/no), diabetes (yes/no), and hypertension (yes/no).

### 2.4. Assessment of Dietary Intakes

A validated and reliable 124-item food frequency questionnaire was used to gather data on participants’ dietary intake [24]. The participants were asked to report the frequency and amount of food and beverage consumption. Frequency of use was reported as “never use,” “daily,” “weekly,” “monthly,” or “annually.” The “daily” or “weekly” intake in the season was asked for seasonal food items. Food intake was entered into Nutritionist IV (First Databank, Hearst Corp., San Bruno, CA, USA), and the average daily intake of macro- and micronutrients was reported. The daily intake was aggregated as the total amount of food in one year and divided by 365 to obtain the average daily intake. The total energy was obtained as the sum of 9 kcal/g for fat, 4 kcal/g for protein, 4 kcal/g for carbohydrates, and 2 kcal/g for dietary fiber. During several sessions, a nutritionist provided the training required to complete the questionnaires accurately. All questionnaires were completed in person by trained medical staff.

### 2.5. HEI Scoring Algorithm

We estimated the HEI for all participants based on FFQ data based on the 2015 update [25]. HEI-2015 components, point values, and standards for scoring are represented in Appendix A. Briefly, the HEI-2015 includes 13 components that reflect the key recommendations of the 2015–2020 dietary guidelines Appendix A. There are two classification approaches: adequacy components represent the food groups, subgroups, and dietary elements recommended for eating. Higher scores reflect higher intakes of these components as higher intakes are desirable. Moderation components represent the food groups and dietary elements for which limited consumption is recommended. For moderation components, higher scores reflect lower intakes as lower intakes are more desirable Appendix A.

Generally, a higher total HEI score signifies a diet that aligns better with dietary recommendations. The HEI components can be deliberated as a set of points, each of which measures compliance/adherence with/to a distinct characteristic of the dietary guidelines/recommendations. Each component (13 items) is allocated a standard for reflecting a maximum score. The components are then summed up to calculate the total HEI-2015 score, with a maximum of 100 points Appendix A. The HEI components are weighted equally as all characteristics are considered similarly important. All components receive a maximum of 10 points. However, some aspects of the diet, such as protein (total protein foods and seafood and plant proteins), are characterized by two components and assigned a maximum of 5 points each. The total HEI score indicates general diet quality, while the component scores, when studied together, represent a pattern of diet quality. The closer a set of food items (i.e., what we collectively drink and eat) aligns with the dietary guidelines, the higher the HEI score. A fundamental aspect of the HEI is that, by using what is termed a density approach, it separates dietary quality from quantity. The components are mainly estimated as a food group amount per 1000 kcal in the total mix of food items. As an exception, the fatty acids component scored as a ratio of unsaturated to SFA [26].

### 2.6. Assessment of Dietary Patterns

We used an exploratory factor analysis (EFA) method to identify dietary patterns (two major components) using the data from the FFQ, organized into 12 major food groups. Absolute values > 0.10 were considered to have a significant role in the components. Small coefficients below this value were suppressed.

### 2.7. Statistical Analysis

The normality of the variables’ distribution was measured by Q-Q normality plots and the Kolmogorov–Smirnov (KS) test. For non-normal distributed data, a log-transformation was performed (log-transferred data were used in all analyses). The HEI (categorized based on the median) was investigated across the following characteristics of participants: age, TAC, MDA, BMI, gender, education, marital status, smoking, alcohol intake, regular physical activity, history of CVD, diabetes, hypertension, and macro- and micronutrient intakes using analyses of independent *t*-test or χ ^2^ test for continuous and categorical variables, respectively. Odds ratios (ORs) and 95% confidence intervals (CI) for obesity/overweight using BMI and BSA as outcomes were estimated using logistic regression in three models. In model A, crude ORs and 95% CIs were reported. In model B, we adjusted for age and gender, and model C was further adjusted for education, alcohol consumption, smoking, history of CVD disease, and total calorie intake. All statistical analyses were performed using SAS^®^ 9.3 (SAS Institute Inc., Cary, NC, USA); all *p*-values were based on two-sided tests. A significance level of 0.05 (two-sided) was considered. Data reported in the descriptive analyses represent mean ± SD.

## 3. Results

### 3.1. Baseline Characteristics and HEI Scores across Various Metabolic Parameters

Table 1 shows demographic characteristics and serum levels of biomarkers across different HEI and BMI groups. According to Table 1, the mean age was 47.9 ± 13.2 years in the participants with an HEI lower than the median and 48.2 ± 12.6 years in those with an HEI higher than the median. There was no significant difference between the two groups of HEI regarding BMI and BSA. Based on Table 1, the participants with higher HEI scores had higher serum HDL-C compared to those with lower HEI scores, which was independent of weight status. There were significant differences between HEI scores between the cases (based on BMI and BSA) and controls.

### 3.2. Comparison of Micro-and Macronutrient Intake According to HEI Scoring

Table 2 compares the dietary intake of macro- and micronutrients based on HEI and BMI classification. The participants (both weight groups combined) with higher HEI scores had significantly lower energy intake. The people with lower HEI scores had a significantly higher intake of total protein, carbohydrate, fat, SFA, PUFA, vitamin A, lutein, vitamin B6, vitamin K, magnesium, manganese, and zinc compared to those with higher HEI scores. Still, these differences can be explained by the significantly higher calorie intake in the group with lower HEI as the group with a lower HEI score had a 25% higher calorie intake. In addition, water consumption in the case group (based on BMI) was significantly lower than the control group by about 5% per day. Contrarily, calorie intake was significantly higher in the case group (based on BMI) than in the control group by about 2.5%.

In addition, we examined the study population based on women and men in HEI and BMI groups. Appendix A shows the comparison (mean ± SD) of the macro-micronutrient intakes of the participants based on gender in HEI and BMI groups. The comparison between gender groups showed relatively similar results compared to the comparisons of the whole population.

Appendix A represents KMO and Bartlett’s Test and Rotated Component Matrix of food groups based on BMI groups. According to EFA, two dietary patterns were determined in the case and control groups based on BMI. In this model, the dietary pattern was similar in the two groups, and it was impossible to determine unique and different major dietary patterns for the two groups (there was no significant difference to determine the dietary pattern based on the food groups). In addition, Appendix A show the component plot in rotated space.

### 3.3. Correlation of HEI and Markers of Inflammation and Oxidative Stress

Table 3 shows the correlation between HEI, serum anti-inflammatory, and antioxidant markers. The bivariate and partial correlation models showed a significant inverse correlation between HEI and IL-4. Similarly, this significant reverse correlation was reported in the age and gender-adjusted and multivariate-adjusted models. The correlations between HEI with TAC and MDA were non-significant. Furthermore, a significant inverse association between HEI and hs-CRP and IL-1β was observed in the multivariate-adjusted model.

### 3.4. Comparison of Macro- and Micronutrient Intakes in Groups According to Inflammatory and Anti-Inflammatory Markers

Table 4 shows the mean ± SD of macro- and micronutrient intakes in different groups (lower and higher than the median) of inflammatory and anti-inflammatory markers. The results of Table 4 show that participants in the group below the median of hs-CRP significantly consumed higher amounts of beta-carotene, vitamin B6, and less iron than those above the median. Moreover, participants in the group below the median of IL-6 consumed significantly more MUFA, PUFA, beta-carotene, vitamins D, B6, and K and less energy, total fat, SFA, and pantothenic acid compared to those in the group above the median. Regarding IL-1β, people in the group below the median consumed significantly more protein, fiber, beta-carotene, alpha-carotene, vitamins B1, B6, copper, and selenium and less energy and total fat compared to those in the group above the median. As for TNF-α, persons below the median consumed significantly more fiber, beta-carotene, and vitamin B6 and less total fat and SFA than those above the median.

On the other hand, the results for IL-10 and IL-4 showed rather the opposite tendency than other markers as these are considered anti-inflammatory markers. However, no significant differences were found between the IL-4 groups. Persons below the median of IL-10 consumed significantly more energy and less fiber, beta-carotene, vitamin B6, folate, and copper than those above the median.

### 3.5. Association of HEI and Overweight/Obesity

Table 5 shows odds ratios (ORs) and 95% CIs for the association between HEI and odds of obesity/overweight. Based on the crude, age- and gender-adjusted, and multivariate models, there was a significant association between HEI (as a continuous variable and a categorical variable) and odds of overweight/obesity across BMI groups; however, the association between HEI and overweight/obesity was not significant in any of the models across BSA groups.

## 4. Discussion

This population-based case-control study’s findings revealed a significant inverse correlation between HEI scores and (anti-)inflammatory markers, including IL-4, IL1β, and hs-CRP. Although no different dietary patterns based on exploratory factor analysis were observed between cases and controls, adherence to dietary recommendations (high HEI scores) was related to lower odds of being overweight/obese. However, overweight/obese participants had significantly lower HEI scores when compared to participants in the normal weight group. However, when we divided the participants into two groups based on the median of HEI, those with HEI scores above the median did not differ significantly in BMI and BSA compared to those with HEI scores below the median.

Although BMI and BSA are valid and reliable indicators of overweight/obesity in large population-based studies, there is a need for further information, such as total body composition and other anthropometric measures, such as waist and hip circumference. Moreover, percentage of body fat and especially body fat distribution are promising markers of obesity [27,28]. In line with our findings, a study analyzing data from 10,930 adults who participated in the Third National Health and Nutrition Examination Survey (a representative sample of the US population and population groups) showed a lower HEI score among individuals with obesity compared to individuals with normal weight [16].

In our study, people with higher HEI scores had significantly higher HDL-C levels, which is in line with previous studies [23,29,30,31]. In a cross-sectional survey by Rashidipour-Fard et al., there was a significant positive association between HDL-C and HEI scores. However, it changed to a non-significant level after adjustment for age, sex, energy intake, and BMI [23]. A higher HEI score represents a higher intake of fruits, vegetables, low-fat dairy products, whole grains, and low-fat meats, which could strongly improve individuals’ blood lipid profiles [23,30]. Following the Iranian traditional dietary pattern has been shown to be significantly associated with an inverse association with dietary inflammatory index (DII) and insulin resistance (IR) in persons with obesity living in the northwest of Iran [32]. A higher intake of different fruits and vegetables in fresh or cooked form, chicken, hydrogenated fats, and red meats was identified as major characteristics in the Iranian dietary pattern [32]. Although there are a few studies on the association of Iranian dietary patterns and inflammatory and metabolic markers, it is evident that a high intake of vegetables and fruits can reduce insulin resistance and pro-inflammatory markers due to the high amount of viscous fiber and their low glycemic index, among other factors [33,34].

Our findings showed that individuals with overweight/obesity, based on BMI, had lower water intake compared to the control group. In accordance with our findings, an interventional study that investigated the effect of pre-meal water drinking on reduction of meal energy intake in middle-aged and older adults reported approximately two kilograms of weight loss and 44% greater weight reduction in the water group compared to the control group [15]. In addition, based on two school-based interventions, increasing water consumption can reduce obesity-related costs and promote public health [35,36]. According to the National Health and Nutrition Examination Survey 2009–2012, adults who were not hydrated sufficiently had higher BMI, although causality and directionality could not be established. Individuals with obesity were proposed more likely to maintain poor hydration or were eating when they were actually thirsty. This suggested that individuals with higher BMI may behave in ways that result in their inadequate rehydration; e.g., they choose to eat foods with high-calorie density instead of those with higher water content. Intake of foods with low water content may contribute to the association between dehydration and elevated BMI. Liquid intake per se could contribute to satiety, and foods higher in fat are generally higher in calorie density, whereas foods high in water or dietary fiber typically have fewer calories per gram and are thus lower in calorie density [35,37].

In our study, people with lower HEI scores had significantly higher dietary intakes of calories and some macro- and micronutrients. Similar results were observed when we made comparisons between men in higher than median HEI with men in lower than median HEI and women in higher than median HEI with women in lower than median HEI. The positive relationship between lower HEI and higher calorie intake has been observed in several studies [38,39,40,41]. Similarly, the Zhu et al. 2016 study suggested that adults with higher diet quality scores had lower energy density and higher eating frequency. More frequent eating was associated with lower dietary energy density, and it was independent of the type of consumed beverages [42]. In the present study, we did not consider the association with eating frequency, but previous findings showed a similar reverse pattern in the diet quality score and dietary energy intake. For example, the results of a one-year cohort study in middle-aged women reported that a lower HEI score was associated with higher calorie intake on weekends, not on weekdays [43]. Similar to our study, the participants with lower healthy dietary patterns received more energy from carbohydrates and fat [43]. Although we did not compare the dietary quality and calorie intake between weekdays and weekends, Jahns et al. found that people who had a higher dietary energy intake had a lower intake of fruits, vegetables, whole grains, and lean meat at weekends, which corresponds to a lower HEI score [43].

There was a negative correlation between HEI and some biomarkers of low-grade chronic inflammation, although not of oxidative stress, and these findings supported our hypothesis that better healthy eating patterns can promote health-related effects via reducing chronic low-grade inflammation [44]. Although we could not observe the same trend for all the inflammatory biomarkers, the reverse correlation was significant for IL-1β, IL-4, and hs-CRP in the crude and adjusted models. Generally speaking, IL-4 is considered an anti-inflammatory cytokine; however, the Binisor et al. study showed a significant association between serum IL-4 and some risk factors of obesity and/or diabetes, such as waist circumstance, hip circumstance, serum lipid profile, glucose, and HBA1c [45]. Along the same line, findings from in vitro studies indicated that chronic activation of macrophages and innate immunity via IL-4 can result in impaired defense against abdominal obesity and diabetes [46]. Although a chronic imbalance of pro-inflammatory mediators (hs-CRP, IL-1β, IL-6, TNF-α) and anti-inflammatory markers (IL-4 and IL-10) has been reported in obesity, the role of each inflammatory mediator is not completely known and needs to be investigated in further analysis, and the cellular function of these cytokines should be defined in the physiopathology of excess white adipose tissue and obesity [45,47].

Confirming our findings, several studies have reported an inverse correlation between the intake of fruits, vegetables, lean meat, and omega-3-rich foods and reduced inflammatory biomarkers [44,48,49,50]. The study by Ford et al. suggested a reverse correlation between the dietary intake of grains and CRP levels [51], and the latter study observed a significant correlation only for CRP levels and HEI scores but not for the other blood biomarkers [52].

In addition, the results of our study showed that participants in the groups below the median of the pro-inflammatory markers (and, conversely, in the groups of anti-inflammatory markers) consumed significantly higher amounts of dietary fiber, beta-carotene, and vitamin B6 and lower amounts of total fat, SFA, and energy compared to the groups above the median. These results are in line with previous findings that showed associations between inflammatory biomarkers and nutrient intakes, such as vitamin B6 [53], carotenoids [54], and dietary fiber [55], and an inverse association with total fat [56] and SFA [57]. The findings highlight that consuming adequate amounts of recommended fruits, vegetables, and whole grains and limiting the intake of saturated fats and refined grains based on the HEI standards may be helpful in controlling chronic inflammation.

The present study could not find a significant association between HEI and antioxidant status biomarkers such as malondialdehyde (MDA). This may result from the insufficient sample size and inter-individual variations for these variables among the participants and due to the fact that antioxidant homeostasis is tightly regulated in blood plasma. In addition, the half-life of MDA in the blood is short, and perhaps lipid peroxidation is not the primary concern in these pathways but rather other routes, such as transcription factors. However, these findings can be supported by the results of Ford et al., 2004 and Monfort-Pires et al., 2014 [51,52]. Investigating F2 isoprostanes and DNA/RNA breakdown products in future studies would be interesting.

Our study showed a significant inverse correlation between HEI and BMI based on pooled findings from the total investigated population, which is in agreement with several population-based studies [16,19,22,58]. A systematic review of dietary quality indices and obesity outcomes suggested that HEI had a significant reverse association with weight gain as well as BMI and waist circumference. Still, there was no data synthesis based on HEI and BSA in the abovementioned systematic review [59]. Adjustment for confounders could not have had a major impact on the data trend, and it can be due to a modified dietary intake among overweight and obese persons who were trying to follow a healthier dietary approach temporarily in order to lose weight (i.e., inverse causality), and the reported dietary intake in a cross-sectional setting cannot represent the long-term intake [59].

This study had a relatively large sample size, and it could provide a valid and reliable piece of evidence to investigate the association between HEI, obesity outcomes, and blood biomarkers of inflammation. We used a valid and reliable FFQ to assess the intake of nutrients. Use of the latest version of HEI (2015) supported measuring both the quality and quantity of dietary intake. We investigated obesity status using two different obesity indicators, and few studies took into account BSA to study HEI and the odds of obesity. BSA has been highlighted to constitute a more appropriate indicator for various CVD [60] and also including obesity [61], as it was argued that BMI alone should be used with caution in heavy individuals [61]. Furthermore, we measured serum inflammatory, lipid peroxidation, and antioxidant biomarkers accompanied by dietary quality index.

One of the limitations of this study was that the study design as a retrospective case-control study has the potential for reporting and recall bias. In population-based nutritional studies, recall bias is a common limitation because of the nature of the questionnaire and the short-term applicability of this technique. However, reporting the dietary intake as FFQ was completed by the trained medical staff to minimize recall bias. Another limitation is that most persons were rather overweight (84.7%) and not obese (15.3%), and BMI differences between cases and controls were thus only moderate. Interestingly, our study also indicated that there were significantly more cases of T2D in the high HEI group compared to the low HEI group. History of T2D was among the baseline variables of our study, and we selected the participants based on their BMI. As a result, this controversial observation could be due to several reasons: (1) representatives: our study subjects were certainly not representative of the entire diabetic population, and this could be a coincidental finding; (2) distribution and sample size: the size of our sample was not sufficient for a proper distribution of diabetic people in HEI groups and perhaps, in this regard, was under-powered; (3) reverse causation: people with diabetes (whose diabetes was diagnosed likely years ago) could be more prudent about their dietary habits and, for this reason, fall into a higher HEI category; i.e., in this case, the disease would be the reason for a higher HEI and not vice versa.

## 5. Conclusions

Although no different dietary patterns were observed between cases and controls, adherence to dietary recommendations (high HEI scores) was related to lower odds of being overweight/obese. In the group with higher HEI, people had higher serum HDL-C independent of weight status. It could be deduced that people with higher HEI scores had improved lipid profiles even though the weight difference was non-significant. There was a significant inverse association between HEI and inflammatory mediators, including IL-1β and hs-CRP. Further prospective studies are required to conclude on the association between HEI, dietary patterns, and odds of obesity based on BSA.

## Figures and Tables

**Table 1 nutrients-14-05127-t001:** Distribution (mean ± SD or number (%)) of socio-demographic and serum variables across HEI and BMI groups.

Characteristics	Mean ± SD or Number (%)						Total (*n* = 1605)
	HEI ^f^			BMI (kg/m^2^)			
	Lower Than Median (*n* = 802)	Higher Than Median (*n* = 803)	*p*-Value ^a^	Overweight/Obese ^b^ BMI ≥ 25 (*n* = 812)	Normal Weight BMI = 17.9 to 24.9 (*n* = 793)	*p*-Value ^a^	
Age (years)	47.9 ± 13.2	48.2 ± 12.6	0.68	47.6 ± 13.0	48.4 ± 12.7	0.26	48.0 ± 12.9
BMI (kg/m^2^)	25.0 ± 3.4	25.1 ± 3.6	0.85	27.8 ± 2.5	22.2 ± 1.7	**<0.01**	25.0 ± 3.5
BSA (m^2^)	1.84 ± 0.19	1.84 ± 0.20	0.91	1.95 ± 0.18	1.72 ± 0.14	**<0.01**	1.84 ± 0.19
HEI	39.3 ± 6.1	61.5 ± 11.0	**<0.01**	49.9 ± 14.0	53.9 ± 14.4	**<0.01**	50.4 ± 14.2
FBS (mg/dL)	93.1 ± 18.8	93.3 ± 20.3	0.82	94.2 ± 20.9	92.1 ± 18.1	**0.03**	93.2 ± 19.6
Insulin (mIU/mL)	7.8 ± 11.7	7.5 ± 5.1	0.79	8.1 ± 11.8	7.2 ± 4.6	**0.04**	7.7 ± 9.0
HbA1C (mmol/mol)	16.7 ± 13.9	17.2 ± 14.2	0.46	18.8 ± 15.5	15.1 ± 12.2	**<0.01**	17.0 ± 14.1
LDL-C (mg/dL)	127.7 ± 33.9	128.8 ± 34.3	0.54	129.9 ± 34.6	126.5 ± 33.4	**0.05**	128.2 ± 34.1
TG (mg/dL)	122.4 ± 75.1	123.2 ± 90.6	0.84	128.5 ± 77.8	116.9 ± 88.1	**<0.01**	122.8 ± 83.2
HDL-C (mg/dL)	53.1 ± 13.7	54.8 ± 14.6	**<0.01**	52.1 ± 13.9	55.9 ± 14.2	**<0.01**	54.0 ± 14.2
Cholesterol (mg/dL)	199.8 ± 38.7	200.8 ± 36.0	0.59	202.4 ± 38.8	198.1 ± 36.2	**0.02**	200.3 ± 37.4
MDA (µmol/L)	3.1 ± 1.9	3.3 ± 1.9	0.30	3.1 ± 1.9	3.3 ± 1.9	**0.03**	3.2 ± 1.9
TAC (mmol/L)	1.6 ± 1.4	1.6 ± 1.2	0.98	1.6 ± 1.3	1.7 ± 1.3	**0.02**	1.6 ± 1.3
Gender			0.34			0.96	
Women	394 (49.1%)	413 (51.4%)		409 (50.3%)	398 (50.2%)		807 (50.3%)
Men	408 (50.9%)	390 (48.6%)		403 (49.7%)	395 (49.8%)		798 (49.7%)
Education			0.34			0.71	
Diploma and Low Literate	544 (67.8%)	526 (65.5%)		545 (67.2%)	525 (66.2%)		1070 (66.6%)
Higher Than Diploma	258 (32.2%)	277 (34.5%)		267 (32.8%)	268 (33.8%)		535 (33.3%)
Smoking status			0.96				
Non-smokers	672 (83.8%)	672 (83.7%)		680 (83.8)	664 (83.8)	0.99	1344 (83.7%)
Alcohol consumption			0.34			0.75	
No	706 (88.0%)	719 (89.5%)		723 (89.1%)	702 (88.5%)		1425 (88.7%)
Marital status			**<0.01**			0.92	
Married	628 (78.3%)	587 (73.1%)		615 (75.7%)	600 (75.6%)		1215 (75.7%)
Single	131 (16.3%)	142 (17.7%)		136 (16.7%)	137 (17.2%)		273 (17.0%)
Other	43 (5.4%)	74 (9.2%)		61 (7.5%)	56 (7.1%)		117 (7.3%)
Regular physical activity			0.24			0.06	
No	614 (76.5%)	594 (73.9%)		628 (77.3%)	580 (73.1%)		1208 (75.2%)
History of T2D			**<0.01**			**<0.01**	
Yes	190 (23.7%)	246 (30.6%)		246 (30.3%)	190 (23.9%)		436 (27.1%)
History of CVD			**0.03**			**<0.01**	
Yes	174 (21.7%)	136 (16.9%)		204 (25.1%)	109 (13.7%)		313 (19.5%)
History of hypertension			0.17			**<0.01**	
Yes	427 (53.2%)	400 (49.8%)		495 (61.0%)	332 (41.8%)		827 (51.5%)

^f^ Categorized based on the HEI median (median = 48.7). ^a^ Independent sample *t*-test was used for comparing continuous variables, and a chi-squared test was used for categorical variables. ^b^ Overweight (84.7%) and obesity (15.3%). SD = standard deviation, BMI = body mass index, BSA = body surface area, HEI = healthy eating index, HbA1C = glycated hemoglobin, LDL-C = low-density lipoprotein-cholesterol, TG = triglycerides, HDL-C = high-density lipoprotein-cholesterol, T2D = type 2 diabetes, CVD = cardiovascular diseases. Significant *p*-values are shown in **bold**.

**Table 2 nutrients-14-05127-t002:** Comparison (mean ± SD) of the macro-/micronutrient intakes of the participants based on HEI and BMI groups.

Variables	Mean ± SD						
	HEI ^f^			BMI			Total (*n* = 1605)
	Lower Than Median (*n* = 802)	Higher Than Median (*n* = 803)	*p*-Value ^a^	Overweight/Obese BMI ≥ 25 (*n* = 812)	Normal Weight BMI = 17.9 to 24.9 (*n* = 793)	*p*-Value ^a^	
Water (g/day)	1490 ± 588.8	1429 ± 556.1	**0.03**	1427 ± 522.2	1492 ± 619.9	**0.02**	1459 ± 573.3
Total energy (kcal/day)	3342 ± 498.2	2513 ± 355.7	**<0.01**	2963 ± 605.5	2890 ± 590.6	**0.01**	2927 ± 599.1
Total protein (g/day)	115.5 ± 42.8	96.1 ± 34.4	**<0.01**	105.3 ± 39.8	106.3 ± 40.2	0.60	105.8 ± 40.0
Carbohydrates (g/day)	437.9 ± 117.4	298.4 ± 64.5	**<0.01**	373.8 ± 115.7	362.3 ± 119.3	**0.05**	368.1 ± 117.6
Total fat (g/day)	125.3 ± 44.4	103.9 ± 29.4	**<0.01**	116.4 ± 40.3	112.8 ± 37.9	0.07	114.6 ± 39.2
SFA (g/day)	47.4 ± 33.9	40.2 ± 24.4	**<0.01**	45.6 ± 33.1	41.9 ± 25.8	**0.01**	43.8 ± 29.8
MUFA (g/day)	29.9 ± 11.5	29.3 ± 9.2	0.25	29.7 ± 10.4	29.5 ± 10.5	0.66	29.6 ± 10.4
PUFA (g/day)	32.5 ± 19.1	27.2 ± 18.0	**<0.01**	29.5 ± 18.6	30.1 ± 18.9	0.52	29.8 ± 18.7
Total fiber (g/day)	44.6 ± 17.6	43.9 ± 16.7	0.43	43.2 ± 17.3	45.2 ± 17.0	**0.02**	44.2 ± 17.2
Soluble fiber (g/day)	6.3 ± 4.0	6.5 ± 4.2	0.27	6.1 ± 3.9	6.7 ± 4.3	**<0.01**	6.4 ± 4.1
Vitamin A (RAE/day)	700.2 ± 356.0	623.8 ± 282.0	**<0.01**	651.8 ± 320.1	627.3 ± 326.4	0.20	662.0 ± 323.2
Beta-carotene (µg/day)	5375 ± 2149	5373 ± 2245	0.98	5264 ± 2202	5487 ± 2188	**0.04**	5374 ± 2197
Alpha-carotene (µg/day)	786.8 ± 412.6	800.5 ± 408.5	0.50	797.6 ± 413.7	789.6 ± 407.7	0.69	793.7 ± 410.6
Lutein (µg/day)	2482 ± 1297	2192 ± 1009	**<0.01**	2350 ± 1184	2324 ± 1157	0.66	2337 ± 1171
Lycopene (µg/day)	5166 ± 2289	5048 ± 2189	0.29	5089 ± 2231	5162 ± 2250	0.74	5107 ± 2240
Vitamin C (mg/day)	152.3 ± 64.2	157.3 ± 59.5	0.10	151.5 ± 58.7	158.2 ± 65.0	**0.03**	154.8 ± 61.9
Calcium (mg/day)	1186 ± 414.1	1202 ± 426.3	0.46	1171 ± 412.2	1218 ± 427.2	**0.02**	1194 ± 420.2
Iron (mg/day)	19.1 ± 0.2	19.0 ± 0.2	0.92	19.0 ± 5.7	18.9 ± 5.6	0.90	19.0 ± 5.7
Vitamin D (IU/day)	2.17 ± 1.61	2.31 ± 1.57	0.07	2.2 ± 1.6	2.2 ± 1.5	0.64	2.24 ± 1.59
Vitamin E (mg/day)	19.1 ± 8.3	18.9 ± 7.4	0.62	19.0 ± 7.8	18.9 ± 7.9	0.83	19.0 ± 7.8
Thiamin (mg/day)	2.22 ± 0.90	2.21 ± 0.86	0.91	2.22 ± 0.88	2.21 ± 0.88	0.77	2.22 ± 0.88
Riboflavin (mg/day)	2.16 ± 0.71	2.18 ± 0.78	0.76	2.17 ± 0.75	2.17 ± 0.75	0.88	2.17 ± 0.75
Niacin (mg/day)	28.8 ± 9.6	30.2 ± 11.1	**<0.01**	29.56 ± 10.45	29.54 ± 10.43	0.96	29.5 ± 10.4
Vitamin B6 (mg/day)	2.56 ± 1.00	2.34 ± 0.89	**<0.01**	2.46 ± 0.95	2.44 ± 0.96	0.64	2.45 ± 0.95
Folate (µg/day)	695.2 ± 225.3	687.1 ± 246.4	0.48	688.7 ± 235.9	693.7 ± 236.4	0.67	691.1 ± 236.1
Vitamin B12 (µg/day)	5.46 ± 3.72	5.61 ± 2.79	0.36	5.52 ± 3.32	5.55 ± 3.25	0.87	5.54 ± 2.29
Biotin (mg/day)	37.6 ± 13.4	38.5 ± 12.9	0.17	38.1 ± 13.1	38.0 ± 13.2	0.86	38.0 ± 13.2
Pantothenic acid (mg/day)	7.5 ± 2.5	7.6 ± 2.6	0.29	7.58 ± 2.59	7.55 ± 2.58	0.83	7.5 ± 2.6
Vitamin K (µg/day)	298.7 ± 159.8	272.5 ± 140.4	**<0.01**	286.2 ± 150.1	285.0 ± 151.8	0.87	285.6 ± 150.9
Magnesium (mg/day)	543.1 ± 160.1	515.7 ± 141.1	**<0.01**	528.0 ± 151.3	530.8 ± 151.6	0.70	529.4 ± 151.4
Zinc (mg/day)	15.4 ± 5.4	14.7 ± 5.1	**<0.01**	15.0 ± 5.2	15.1 ± 5.2	0.83	15.1 ± 5.2
Copper (µg/day)	2.6 ± 1.1	2.6 ± 1.2	0.77	2.68 ± 1.22	2.65 ± 1.20	0.64	2.6 ± 1.2
Manganese (mg/day)	9.0 ± 3.2	8.6 ± 3.3	**0.04**	8.82 ± 3.29	8.85 ± 3.28	0.87	8.8 ± 3.2
Selenium (µg/day)	129.4 ± 51.4	129.0 ± 47.0	0.89	126.6 ± 45.6	131.9 ± 52.6	**0.03**	129.2 ± 49.2

^f^ Categorized based on the HEI median = 48.7. ^a^ Independent sample *t*-test was used for comparing nutrients intakes. BMI = body mass index, HEI = healthy eating index, SFA= saturated fatty acids, MUFA = monounsaturated fatty acids, PUFA = polyunsaturated fatty acids. Significant *p*-values are shown in **bold**.

**Table 3 nutrients-14-05127-t003:** Pearson and partial correlation coefficients between HEI and serum levels of inflammatory and antioxidant biomarkers (*n* = 1605).

Biomarkers	CC	*p*-Value ^a^	CC ^f^	*p*-Value ^b^	CC ^f^	*p*-Value ^c^
TAC	0.013	0.60	0.013	0.60	−0.039	0.12
MDA	−0.016	0.51	−0.016	0.52	−0.013	0.60
hs-CRP	0.007	0.78	0.007	0.78	−0.063	**<0.01**
IL-6	−0.018	0.48	−0.018	0.48	0.022	0.37
IL-1β	−0.076	**<0.01**	−0.076	**<0.01**	−0.054	**0.03**
TNF-α	0.005	0.83	0.005	0.83	0.027	0.27
IL-10	0.073	**<0.01**	0.074	**<0.01**	0.011	0.65
IL-4	−0.074	**<0.01**	−0.074	**<0.01**	−0.069	**<0.01**
HbA1C	−0.006	0.80	−0.006	0.80	0.009	0.72
Insulin	0.018	0.47	0.018	0.46	−0.013	0.60

^a^ Bivariate correlation. ^b^ Controlling for age and gender. ^c^ Additionally controlling for education, total energy intake, BMI, marital status, smoking, regular physical activity, and disease history, including hypertension, CVD, and type 2 diabetes. ^f^ Partial correlation models were used to control potential cofounders. CC = correlation coefficient, TAC = total antioxidant capacity, MDA = malondialdehyde, BMI = body mass index, HOMA-IR = homeostatic model assessment for insulin resistance, hs-CRP = high-sensitivity C-reactive protein, IL = interleukin, HBA1C = hemoglobin A1c (glycated hemoglobin), T2D = type 2 diabetes, CVD = cardiovascular diseases. Significant *p*-values are shown in **bold**.

**Table 4 nutrients-14-05127-t004:** Comparison (mean ± SD) of macro- and micronutrient intakes in various groups (lower and higher than the median) according to inflammatory markers ^a^.

Variables	Mean ± SD																	
	hs-CRP			IL-6			IL-1β			TNF-α			IL-10			IL-4		
	< Median (*n* = 817)	≥ Median (*n* = 788)	*p*-Value	< Median (*n* = 816)	≥ Median *(n* = 789)	*p*-Value	< Median (*n* = 816)	≥ Median (*n* = 789)	*p*-Value	< Median (*n* = 813)	≥ Median (*n* = 792)	*p*-Value	< Median (*n* = 815)	≥ Median (*n* = 790)	*p*-Value	< Median (*n* = 805)	≥ Median (*n* = 800)	*p*-Value
Water (g/day)	1466 ± 594.9	1452 ± 550.3	0.633	1440 ± 569.7	1479 ± 576.7	0.183	1479 ± 591.7	1438 ± 553.2	0.152	1473 ± 579.4	1445 ± 566.9	0.326	1460 ± 587.2	1458 ± 558.9	0.953	1468 ± 589.9	1450 ± 556.4	0.539
Total energy (kcal/day)	2926 ± 613.5	2928 ± 584.2	0.925	2892 ± 580.4	2963 ± 616.2	**0.018**	2894 ± 569.3	2961 ± 627.0	**0.025**	2910 ± 591.5	2945 ± 606.7	0.246	2957 ± 610.7	2896 ± 585.5	**0.044**	2904 ± 600.4	2950 ± 597.3	0.117
Total protein (g/day)	104.1 ± 39.5	107.5 ± 40.5	0.099	107.0 ± 40.0	104.5 ± 40.0	0.207	108.1 ± 40.6	103.3 ± 39.3	**0.016**	106.8 ± 40.2	104.7 ± 39.8	0.277	106.1 ± 40.6	105.4 ± 39.5	0.704	105.2 ± 41.1	106.3 ± 38.9	0.599
Carbohydrates (g/day)	368.4 ± 120.1	367.7 ± 115.0	0.902	363.8 ± 113.9	372.5 ± 121.2	0.138	363.2 ± 112.5	373.2 ± 122.5	0.088	369.7 ± 121.1	366.4 ± 113.9	0.568	371.6 ± 115.8	364.4 ± 119.3	0.222	363.9 ± 117.4	372.3 ± 117.7	0.156
Total fat (g/day)	115.0 ± 39.6	114.2 ± 38.6	0.670	112.1 ± 36.6	117.2 ± 41.5	**0.009**	112.1 ± 36.1	117.2 ± 41.9	**0.008**	111.5 ± 35.6	117.8 ± 42.3	**0.001**	116.2 ± 40.0	113.0 ± 38.2	0.106	114.1 ± 38.3	115.1 ± 40.0	0.597
SFA (g/day)	43.1 ± 30.0	44.5 ± 29.5	0.362	41.9 ± 27.4	45.7 ± 32.0	**0.010**	42.6 ± 29.2	45.0 ± 30.3	0.101	41.0 ± 25.5	46.6 ± 33.3	**<0.001**	44.8 ± 30.1	42.6 ± 29.4	0.139	43.6 ± 29.6	43.9 ± 29.9	0.822
MUFA (g/day)	30.1 ± 10.5	29.1 ± 10.3	0.061	30.2 ± 10.0	29.0 ± 10.8	**0.020**	29.9 ± 10.1	29.2 ± 10.8	0.176	30.0 ± 10.0	29.2 ± 10.8	0.149	29.3 ± 11.0	29.9 ± 9.8	0.322	30.1 ± 10.2	29.1 ± 10.6	0.078
PUFA (g/day)	30.4 ± 19.0	29.2 ± 18.4	0.218	30.4 ± 19.4	29.2 ± 18.0	0.227	30.1 ± 18.6	29.5 ± 18.8	0.570	30.3 ± 18.9	29.4 ± 18.5	0.344	29.6 ± 18.3	30.0 ± 19.1	0.670	30.2 ± 18.7	29.4 ± 18.7	0.429
Total fiber (g/day)	45.0 ± 17.5	43.4 ± 16.8	0.075	44.9 ± 17.5	43.5 ± 16.8	0.117	45.9 ± 17.9	42.5 ± 16.2	**<0.001**	45.2 ± 17.7	43.2 ± 16.5	**0.019**	43.4 ± 16.5	45.1 ± 17.8	**0.045**	44.9 ± 17.8	43.5 ± 16.5	0.087
Soluble fiber (g/day)	0.64 ± 0.42	0.64 ± 0.41	0.670	0.63 ± 0.41	0.65 ± 0.41	0.234	0.64 ± 0.40	0.64 ± 0.42	0.739	0.63 ± 0.41	0.65 ± 0.42	0.411	0.65 ± 0.41	0.63 ± 0.42	0.284	0.63 ± 0.41	0.65 ± 0.41	0.569
Vitamin A (RAE/day)	676.2 ± 348.1	647.2 ± 294.8	0.072	665.6 ± 347.6	658.2 ± 296.1	0.647	671.7 ± 342.5	651.9 ± 301.9	0.222	674.9 ± 344.7	648.7 ± 299.3	0.104	656.8 ± 317.8	667.3 ± 328.9	0.517	670.7 ± 327.9	653.2 ± 318.5	0.227
Beta-carotene (µg/day)	5496 ± 2263	5247 ± 2120	**0.023**	5555 ± 2263	5186 ± 212	**<0.001**	5589 ± 2198	5152 ± 2175	**<0.001**	5540 ± 2245	5204 ± 2134	**0.002**	5182 ± 2136	5572 ± 2242	**<0.001**	5433 ± 2191	5314 ± 2203	0.280
Alpha-carotene (µg/day)	796.6 ± 421.6	790.6 ± 408.8	0.767	804.3 ± 422.9	782.7 ± 379.5	0.293	820.4 ± 428.7	765.9 ± 389.4	**0.008**	802.5 ± 421.1	784.6 ± 399.6	0.383	794.7 ± 403.5	792.5 ± 418.0	0.914	799.8 ± 419.7	787.9 ± 401.4	0.549
Lutein (µg/day)	2379 ± 1229	2294 ± 1106	0.146	2352 ± 1124	2322 ± 1217	0.610	2304 ± 1107	2371 ± 1233	0.248	2333 ± 1111	2341 ± 1229	0.899	2367 ± 1229	2306 ± 1107	0.291	2377 ± 1191	2297 ± 1149	0.173
Lycopene (µg/day)	5108 ± 2227	5107 ± 2254	0.993	5104 ± 2230	5110 ± 2250	0.955	5151 ± 2237	5063 ± 2243	0.444	5009 ± 2188	5208 ± 2288	0.075	5112 ± 2269	5102 ± 2210	0.926	5115 ± 2210	5099 ± 2270	0.888
Vitamin C (mg/day)	156.5 ± 58.1	153.0 ± 65.7	0.252	156.9 ± 59.5	152.6 ± 64.3	0.173	155.3 ± 57.9	154.3 ± 65.8	0.763	155.3 ± 59.0	154.3 ± 64.8	0.735	154.1 ± 65.9	155.5 ± 57.6	0.633	156.3 ± 62.1	153.3 ± 61.7	0.338
Calcium (mg/day)	1204 ± 419.1	1184 ± 421.5	0.326	1205 ± 412.2	1183 ± 428.3	0.275	1211 ± 420.6	1176 ± 419.3	0.096	1192 ± 414.3	1196 ± 426.4	0.857	1183 ± 417.9	1206 ± 422.5	0.271	1181 ± 412.5	1207 ± 427.6	0.222
Iron (mg/day)	18.7 ± 5.5	19.3 ± 5.8	**0.024**	18.9 ± 5.6	19.0 ± 5.7	0.683	18.8 ± 5.6	19.1 ± 5.7	0.369	18.9 ± 5.6	19.0 ± 5.7	0.645	19.1 ± 5.6	18.9 ± 5.7	0.153	19.1 ± 5.8	18.9 ± 5.5	0.492
Vitamin D (IU/day)	2.24 ± 1.59	2.24 ± 1.59	0.999	2.33 ± 1.61	2.14 ± 1.57	**0.022**	2.25 ± 1.58	2.23 ± 1.60	0.805	2.29 ± 1.58	2.18 ± 1.60	0.166	2.19 ± 1.57	2.29 ± 1.61	0.196	2.27 ± 1.57	2.20 ± 1.61	0.384
Vitamin E (mg/day)	19.3 ± 7.9	18.6 ± 7.7	0.056	19.0 ± 7.5	19.0 ± 8.1	0.983	18.8 ± 7.4	19.1 ± 8.3	0.442	19.2 ± 7.5	18.7 ± 8.2	0.165	18.6 ± 7.9	19.3 ± 7.8	0.078	18.9 ± 7.3	19.0 ± 8.4	0.982
Thiamin (mg/day)	2.23 ± 0.86	2.21 ± 0.90	0.638	2.26 ± 0.92	2.17 ± 0.83	0.051	2.27 ± 0.92	2.16 ± 0.82	**0.011**	2.23 ± 0.91	2.20 ± 0.85	0.547	2.18 ± 0.74	2.25 ± 0.91	0.113	2.22 ± 0.83	2.21 ± 0.92	0.932
Riboflavin (mg/day)	2.15 ± 0.77	2.19 ± 0.73	0.300	2.19 ± 0.78	2.15 ± 0.71	0.282	2.17 ± 0.77	2.17 ± 0.72	0.814	2.20 ± 0.80	2.14 ± 0.70	0.130	2.17 ± 0.73	2.17 ± 0.76	0.888	2.16 ± 0.73	0.218 ± 0.76	0.572
Niacin (mg/day)	29.5 ± 10.5	29.5 ± 10.3	0.905	29.6 ± 10.8	29.4 ± 10.0	0.820	29.7 ± 10.7	29.3 ± 10.1	0.463	29.3 ± 10.4	29.7 ± 10.3	0.444	29.9 ± 10.3	29.1 ± 10.5	0.148	29.6 ± 10.4	29.4 ± 10.4	0.750
Vitamin B6 (mg/day)	2.54 ± 1.01	2.36 ± 0.88	**<0.001**	2.56 ± 1.02	2.34 ± 0.86	**<0.001**	2.54 ± 0.97	2.36 ± 0.92	**<0.001**	2.60 ± 1.03	2.30 ± 0.84	**<0.001**	2.36 ± 0.87	2.55 ± 1.02	**<0.001**	2.49 ± 0.96	2.42 ± 0.94	0.152
Folate (µg/day)	696.6 ± 234.4	685.4 ± 237.8	0.343	696.7 ± 226.6	685.4 ± 245.5	0.337	699.6 ± 221.5	682.4 ± 250.0	0.144	692.0 ± 218.0	690.2 ± 253.4	0.879	678.6 ± 245.2	704.1 ± 225.6	**0.030**	689.5 ± 235.9	692.8 ± 236.3	0.777
Vitamin B12 (µg/day)	5.50 ± 2.91	5.57 ± 3.64	0.673	5.48 ± 3.11	5.59 ± 3.46	0.502	5.56 ± 2.85	5.52 ± 3.68	0.823	5.49 ± 3.17	5.59 ± 3.40	0.548	5.48 ± 3.42	5.59 ± 3.14	0.488	5.53 ± 3.45	5.55 ± 3.11	0.899
Biotin (mg/day)	37.9 ± 13.0	38.2 ± 13.3	0.676	38.6 ± 13.1	37.4 ± 13.2	0.078	38.6 ± 13.1	37.5 ± 13.2	0.086	38.2 ± 12.9	37.8 ± 13.5	0.526	37.8 ± 13.1	38.2 ± 13.3	0.572	37.9 ± 13.3	38.1 ± 13.0	0.801
Pantothenic acid (mg/day)	7.59 ± 2.52	7.54 ± 2.65	0.674	7.41 ± 2.38	7.73 ± 2.78	**0.013**	7.55 ± 2.49	7.58 ± 2.69	0.793	7.62 ± 2.49	7.50 ± 2.68	0.354	7.52 ± 2.62	7.61 ± 2.56	0.464	7.62 ± 2.58	7.51 ± 2.60	0.390
Vitamin K (µg/day)	290.5 ± 153.3	280.5 ± 148.1	0.184	295.0 ± 151.2	275.9 ± 150.1	**0.011**	287.8 ± 150.6	283.6 ± 151.3	0.545	287.3 ± 152.4	283.8 ± 149.3	0.647	285.8 ± 153.2	285.3 ± 148.6	0.945	286.4 ± 149.3	284.8 ± 152.6	0.836
Magnesium (mg/day)	528.9 ± 150.6	529.8 ± 152.4	0.905	526.2 ± 148.3	532.6 ± 154.5	0.399	524.3 ± 145.9	534.6 ± 156.8	0.171	530.7 ± 151.2	528.0 ± 151.7	0.719	529.6 ± 157.2	529.1 ± 145.3	0.951	524.6 ± 149.3	534.1 ± 153.5	0.211
Zinc (mg/day)	15.18 ± 5.38	15.05 ± 5.09	0.645	15.29 ± 5.61	14.94 ± 4.81	0.184	15.13 ± 5.13	15.10 ± 5.34	0.898	15.16 ± 5.46	15.07 ± 5.00	0.719	15.25 ± 5.06	14.89 ± 5.42	0.306	15.14 ± 5.32	15.09 ± 5.16	0.866
Copper (µg/day)	2.70 ± 1.19	2.63 ± 1.23	0.252	2.71 ± 1.18	2.62 ± 1.24	0.133	2.74 ± 1.20	2.59 ± 1.22	**0.014**	2.71 ± 1.15	2.62 ± 1.26	0.144	2.59 ± 1.20	2.75 ± 1.21	**0.011**	2.70 ± 1.22	2.63 ± 1.19	0.252
Manganese (mg/day)	8.78 ± 3.22	8.89 ± 3.35	0.518	8.79 ± 3.25	8.88 ± 3.33	0.526	8.82 ± 3.23	8.85 ± 3.34	0.882	8.81 ± 3.18	8.86 ± 3.40	0.781	8.82 ± 3.33	8.86 ± 3.24	0.797	8.76 ± 3.29	8.92 ± 3.28	0.342
Selenium (µg/day)	127.3 ± 47.6	131.1 ± 50.8	0.129	130.7 ± 47.7	127.6 ± 50.7	0.216	126.6 ± 46.3	131.8 ± 52.0	**0.033**	130.0 ± 48.7	128.3 ± 49.8	0.485	128.7 ± 50.0	129.6 ± 48.5	0.722	131.3 ± 51.3	127.0 ± 47.0	0.081

^a^ Independent sample *t*-test was used for comparing nutrients intakes across inflammatory markers groups. SFA = saturated fatty acids, MUFA = monounsaturated fatty acids, PUFA = polyunsaturated fatty acids, hs-CRP = high-sensitivity C-reactive protein, IL = interleukin. Significant *p*-values are shown in **bold**.

**Table 5 nutrients-14-05127-t005:** Multivariable logistic regression (ORs and 95% CI) of the associations between HEI (as a continuous and categorical variable) and obesity/overweight categories * (*n* = 1605).

Models	HEI (Categorized Based on the Median: 48.7)					HEI (Continuous)			
	HEI ≤ 48.7	HEI > 48.7							
		BMI	*p*-Value	BSA	*p*-Value	BMI	*p*-Value	BSA	*p*-Value
**Model A**	1 (ref.)	0.822 (0.675, 0.999)	**0.05**	0.856 (0.704, 1.042)	0.12	0.998 (0.998, 0.999)	**0.02**	0.997 (0.993, 1.001)	0.14
**Model B**	1 (ref.)	0.821 (0.674, 0.999)	**0.05**	0.848 (0.687, 1.046)	0.12	0.994 (0.988, 0.999)	**0.02**	0.998 (0.993, 1.002)	0.31
**Model C**	1 (ref.)	0.801 (0.658, 0.977)	**0.02**	0.838 (0.678, 1.036)	0.10	0.993 (0.988, 0.999)	**0.01**	0.998 (0.993, 1.002)	0.29

Model A: crude ORs. Model B: adjusted for age and gender. Model C: model B+ additionally adjusted for education, marital status, alcohol consumption, smoking, and history of baseline disease, including type 2 diabetes, hypertension, and cardiovascular diseases. * BMI and BSA as dependent variables were entered as categorical variables (1 = overweight/obese or 0= normal weight). HEI = healthy eating index, BMI = body mass index, OR = odds ratio, CI = confidence interval, ref = reference, BSA = body surface area. Significant *p*-values are shown in **bold.**

## Data Availability

The data presented in this study are available on request from the corresponding author. The data are not publicly available due to our institute/university rules and laws.

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
