# Peer review of "A Higher Healthy Eating Index Is Associated with Decreased Markers of Inflammation and Lower Odds for Being Overweight/Obese Based on a Case-Control Study"

_nutrients, 2022, doi:10.3390/nu14235127_

Round 1

Reviewer 1 Report

Although this clinical study is well powered by a large number of participants, the finding that healthy eating (shown by an index score) reduces the likelihood of obesity and inflammation is not new and very much expected.

Major points:

At the end of the introduction it is stated that the population was Iranian (this is lacking in the abstract), but even if that is a rare investigation, I do not really see a scientific novelty.

A lack in the methodical design is that there is no differentiation btw overweight and obese. For some analytical aspects a separation (or exclusion of overweights with BMI>25<30)  could be useful.

The HEI index itself may correlate with high HDL, reduced obesity and some inflammatory parameters in this study, but simlar results were shown by other studies with more detailed in depth investigations available, one is also cited.

Further there is one controverse observation in Table 1 that is not explained. The study showed that there were significantly more cases of Diabetes (T2??) in the high HEI index group compared to the low HEI group. This is opposite to their (expected) observation with normal and higher BMI separation. Neither results nor discussion focusses on that.

Table 2 shows comparative micro and macro nutrients, but the outcome is that the found differences (with some higher nutrient contents in the low HEI group) most probably occur due to the higher caloric intake of the low HEI group. Not really a new finding. A gender-based separation could be of interest here, or in the supplements, but is not shown.

Another requirement for a positive consideration of this study could be an analysis why these study participants show certain cytokine response/immune profiles. The immunological response of diet "responders" vs. "non-responders" could be of high interest for analysis. If such investigations are added, the outlined study could be considered for a reevaluation.

However, with the outlined content, the manuscript is possibly not appropriate for Nutrients in its current form.

Minor points:

Methods

line 157 total soluble fiber content? insoluble fibers do not provide energy as I am aware..

Discussion

line 305-307: In terms of cited content, the last part of that sentence does not seem to match to the introduced citation(s).

line 355:  in future

Author Response

Dear reviewers, thank you for your valuable comments. We went through the comments and the manuscript again and corrected the relevant points as much as we could.

Response to reviewer 1:

Although this clinical study is well powered by a large number of participants, the finding that healthy eating (shown by an index score) reduces the likelihood of obesity and inflammation is not new and very much expected. Major points:

At the end of the introduction, it is stated that the population was Iranian (this is lacking in the abstract), but even if that is a rare investigation, I do not really see a scientific novelty.

Reply: We understand the reviewer’s concerns. We have added “Iranian” (Line 17) to the abstract as the reviewer asked us. This case-control study was conducted in Iranian people to observe the association between HEI and odds ratio of obesity (based on two criteria: BMI and BSA – body surface area) and inflammation markers in obesity setting. In addition to studying both dietary patterns and metabolic endpoints in a relatively large cohort, which is not often done together, we have further added BSA to the more conventional approach, namely using BMI. This allowed us to make a comparison between different methods of overweight/obesity classification, which is somewhat rarely carried out and is possibly the main innovative aspect of this study. We have further tried to highlight this aspect better in the introduction and discussion.

A lack in the methodical design is that there is no differentiation btw overweight and obese. For some analytical aspects a separation (or exclusion of overweights with BMI>25<30) could be useful.

Reply: We agree with the reviewer. Indeed, separating participants with obesity and overweight would have added interesting insights to the article. However, as mentioned in the discussion section (Lines 412-413), one of the limitations of our study is that most persons were rather being overweight (84.7%) and not having obesity (15.3%), and BMI differences between cases and controls were thus relatively moderate. Therefore, Future studies focusing on only one of the groups, either obese or overweight, may be superior approach to reveal relationships between eating patterns and weight-related outcomes.

The HEI index itself may correlate with high HDL, reduced obesity, and some inflammatory parameters in this study, but similar results were shown by other studies with more detailed in-depth investigations available, one is also cited.

Reply: In the present study, we not only investigated the correlation between HEI with some key biochemical markers but also focused on more detailed aspects of dietary patterns and the classification of overweight and obesity based on two criteria (BMI and BSA). There are some relevant studies that have investigated the association between diet quality with some cardiovascular risk factors, including lipoproteins and inflammatory markers (Monfort-Pires et al., 2014; Millar et al., 2021; Lavoi et al., 2013; Asadi et al., 2020; Nicklas et al., 2012). However, the diet quality in the above mentioned studies is one aspect of the parameters investigated in the present study, while the markers of inflammation and the risk factors related to disease/disorder differ compared to the present study, and none of them has been including BSA. As a limitation of our study, we confirm that the quality of this case-control study can be improved by adding more precise markers such as genomics and metabolomics data in order to investigate the antioxidant status and inflammatory markers in people with overweight and obesity, but this is typically very costly in large studies. Our study can be regarded as a feasibility study for larger population-based studies to test the effect of traditional dietary habits on diet quality and objective metabolic variables obtained from blood measurements.

Further there is one controverse observation in Table 1 that is not explained. The study showed that there were significantly more cases of Diabetes (T2??) in the high HEI index group compared to the low HEI group. This is opposite to their (expected) observation with normal and higher BMI separation. Neither results nor discussion focusses on that.

Reply: This is well observed indeed. The history of type 2 diabetes and other cardiometabolic diseases were among the baseline variables. Since the duration of these diseases is unknown, and our questionnaire (FFQ) only examined the last year's intake, the diet status of people with disorders before this period cannot be determined. However, in the final models, we considered them as confounders and controlled/adjusted for them in the models (within history of baseline diseases including T2D).

Table 2 shows comparative micro and macro nutrients, but the outcome is that the found differences (with some higher nutrient contents in the low HEI group) most probably occur due to the higher caloric intake of the low HEI group. Not really a new finding. A gender-based separation could be of interest here, or in the supplements, but is not shown.

Reply: We agree with the reviewer. We added a supplementary table 2, which is a gender-based analysis. In this table, we compared the macro-micronutrient intakes of the participants based on HEI, BMI, and gender groups. Although the results were similar to comparisons of the entire study population, they show more details about how intakes were distributed among men and women.

Another requirement for a positive consideration of this study could be an analysis why these study participants show certain cytokine response/immune profiles. The immunological response of diet "responders" vs. "non-responders" could be of high interest for analysis. If such investigations are added, the outlined study could be considered for a reevaluation.

However, with the outlined content, the manuscript is possibly not appropriate for Nutrients in its current form.

Reply: We agree with the reviewer. We have added a new section to the article (results): 3.4. Comparison of macro and micronutrients intakes according to groups of (anti-)inflammatory markers. In this section, we compared the macro- and micronutrient intakes according to groups of inflammatory and anti-inflammatory markers (lower vs. higher than the median). Results are presented now in table 4, and we have added a related section in the discussion (lines 370-378).

Minor points: Methods

line 157 total soluble fiber content? insoluble fibers do not provide energy as I am aware.

Reply: We have rephrased the sentence for more clarity (line 158). We considered this reference for our calculation: (Factors for dietary fiber vary widely and are not dependent on the method. Energy values for dietary fiber are: 0 kJ/g (0 kcal/g) for non-fermentable fiber; 0 to 17 kJ/g (0 to 4.0 kcal/g) for fermentable fiber; and 0 to 8 kJ/g (0 to 1.9 kcal/g) for commonly eaten foods that contain a mixture of fermentable (assumed to be on average 70 percent of the total) and non-fermentable fiber, so that an average of 2 kcal/g is usually applied (https://www.fao.org/3/y5022e/y5022e04.htm)).

Discussion line 305-307: In terms of cited content, the last part of that sentence does not seem to match to the introduced citation(s).

Reply: We have edited the cited content.

line 355:  in future

Reply: It has been corrected.

Reviewer 2 Report

Comments

This study is shown the relationship between Healthy Eating Index Score and overweight/obese (BMI and BSA).

It is also an interesting and clear results of examining the application of HEI to the evaluation of Iranian diet based on previous studies reports.

 In Discussion section, you mentioned as follows.

L291

In our study, people with higher HEI scores had significantly higher HDL-C levels, which is in line with the previous studies [23,29-31]. In a cross-sectional survey by Rashidipour-Fard et al., there was a significant positive association between HDL-C and HEI scores. Still, it changed to a non-significant level after adjustment for age, sex, energy intake, and BMI [23]. A higher HEI score represents a higher intake of fruits, vegetables, low-fat dairy products, whole grains, and low-fat meats, which could strongly improve the individuals' blood lipid profile [23,30].

Based on the above contents, could you please include the contents on the following into the Discussion?

Considering Iranian historical food culture and dietary patterns, are there any points to keep in mind when using HEI to evaluate Iranian diet?

Author Response

Dear reviewers, thank you for your valuable comments. We went through the comments and the manuscript again and corrected the relevant points as much as we could.

This study is shown the relationship between Healthy Eating Index Score and overweight/obese (BMI and BSA). It is also an interesting and clear results of examining the application of HEI to the evaluation of Iranian diet based on previous studies reports. In Discussion section, you mentioned as follows.

L291: In our study, people with higher HEI scores had significantly higher HDL-C levels, which is in line with the previous studies [23,29-31]. In a cross-sectional survey by Rashidipour-Fard et al., there was a significant positive association between HDL-C and HEI scores. Still, it changed to a non-significant level after adjustment for age, sex, energy intake, and BMI [23]. A higher HEI score represents a higher intake of fruits, vegetables, low-fat dairy products, whole grains, and low-fat meats, which could strongly improve the individuals' blood lipid profile [23,30].

Based on the above contents, could you please include the contents on the following into the Discussion? Considering Iranian historical food culture and dietary patterns, are there any points to keep in mind when using HEI to evaluate Iranian diet?

Reply: We thank the reviewer for such promising words; much appreciated.

We have now added some lines regarding Iranian food culture and dietary patterns according to the available literature (Lines 311-319).

Round 2

Reviewer 1 Report

Minor points:

The study showed that there were significantly more cases of T2D in the high HEI index group compared to the low HEI group.

You still did not clearly try to answer this - still controverse observation. One would expect high HEI with less cases of T2D when compared to the low HEI group. This is the opposite here.

Line 330-333

According to National Health and Nutrition Examination Survey 2009-2012, the adults who were not hydrated sufficiently had higher BMI due to lower meal calorie intake and reduced appetite [35,37].

I still don´t understand that paragraph. How can they have a higher BMI, so obviously more weight, DUE TO a lower meal calorie intake and a reduced appetite? Wouldn´t it be the opposite? Please clarify finally.

I could not find the new Table 2 in the supplements. Maybe it was forgotten in the uploading process.

Author Response

Dear reviewer, we appreciate your comments. We have reviewed the comments and the manuscript again and corrected the relevant points.

Minor points:

- The study showed that there were significantly more cases of T2D in the high HEI index group compared to the low HEI group.

You still did not clearly try to answer this - still controverse observation. One would expect high HEI with less cases of T2D when compared to the low HEI group. This is the opposite here.

Reply: We agree that it is indeed a controversial point. Perhaps our previous explanations were not elaborated enough indeed:

In the present study, the history of type 2 diabetes was among the baseline variables of our study, and we selected the participants based on their BMI. As a result, the controversial observation can be due to several reasons: 1) Representatives: our study subjects are certainly not representative of the entire diabetic population, and this could be a coincidential finging, 2) Distribution and sample size: the size of our sample is not enough for a proper distribution of diabetic people in HEI groups, and perhaps in this regard under-powered, 3) Reverse causation: people with diabetes (whose diabetes was diagnosed maybe years ago) could be more prudent about their dietary habits and for this reason fall into a higher HEI category, i.e. in this case the disease is the reason for a higher HEI, not vice versa.

Since we cannot be certain on these above aspects in regard to baseline variables in case-control studies, we have to control/adjust their effect in statistical models, and this is what has been done in the final models, which included diseases such as type 2 diabetes.

We have added a paragraph to the study's limitations and explained this observation further (lines 424-434).

- According to National Health and Nutrition Examination Survey 2009-2012, the adults who were not hydrated sufficiently had higher BMI due to lower meal calorie intake and reduced appetite [35, 37]. I still don´t understand that paragraph. How can they have a higher BMI, so obviously more weight, DUE TO a lower meal calorie intake and a reduced appetite? Wouldn´t it be the opposite? Please clarify finally.

Reply: We apologise, perhaps our phrasing was not very accurate. We have modified the relevant part for more clarity (lines 330-338). It is mainly the aspect of satiety that we tried to emphasize.

- I could not find the new Table 2 in the supplements. Maybe it was forgotten in the uploading process.

Reply: We appreciate your comment; this may have been due to a mistake during uploading, it should be visible now.